# Non-G Base Tetrads

**DOI:** 10.3390/molecules27165287

**Published:** 2022-08-19

**Authors:** Núria Escaja, Bartomeu Mir, Miguel Garavís, Carlos González

**Affiliations:** 1Organic Chemistry Section, Inorganic and Organic Chemistry Department, University of Barcelona, Martí i Franquès 1–11, 08028 Barcelona, Spain; 2Institute of Biomedicine, University of Barcelona, Av. Diagonal 645, 08028 Barcelona, Spain; 3Instituto de Química Física ‘Rocasolano’, CSIC, Serrano 119, 28006 Madrid, Spain

**Keywords:** DNA structure, RNA structure, quadruplex, tetrad

## Abstract

Tetrads (or quartets) are arrangements of four nucleobases commonly involved in the stability of four-stranded nucleic acids structures. Four-stranded or quadruplex structures have attracted enormous attention in the last few years, being the most extensively studied guanine quadruplex (G-quadruplex). Consequently, the G-tetrad is the most common and well-known tetrad. However, this is not the only possible arrangement of four nucleobases. A number of tetrads formed by the different nucleobases have been observed in experimental structures. In most cases, these tetrads occur in the context of G-quadruplex structures, either inserted between G-quartets, or as capping elements at the sides of the G-quadruplex core. In other cases, however, non-G tetrads are found in more unusual four stranded structures, such as i-motifs, or different types of peculiar fold-back structures. In this report, we review the diversity of these non-canonical tetrads, and the structural context in which they have been found.

## 1. Introduction

Interest in four-stranded nucleic acids structures has been constantly increasing in the last two decades. Although not always the case, four-stranded structures are usually stabilized by arrangements of four nucleobases commonly named as tetrads or quartets. The most extensively studied four-stranded structure is the G-quadruplex [1], the importance of which, in recent times, has been driven by its proven biological relevance and promising therapeutic applications [2,3,4,5].

G-quadruplexes exhibit wide structural diversity depending on strands orientation (parallel, antiparallel and hybrid structures) and connecting loop topologies (diagonal, lateral and propeller loops) [6,7,8]. Despite their structural variety, all G-quadruplexes share the same common element: the so-called G-tetrad. G-tetrads are platforms of four guanine nucleobases interacting through their Hoogsteen and Watson–Crick sides, following either a clockwise, or an anticlockwise arrangement (Figure 1A). These tetrads are stabilized by eight hydrogen bonds, and by electrostatic interactions with cations located between two consecutive tetrads. The cation, usually monovalent, compensates the negatively charged oxygens in the center of the tetrad. Among the different monovalent cations, K^+^ is the most stabilizing, since its atomic radius fits very well in the central position of two consecutive tetrads. G-tetrads are planar, allowing an efficient stacking of multiple layers, and giving rise to extremely thermostable structures.

Although the G-quartet is the most common and well-studied tetrad, this is not the only possible arrangement of four nucleobases. In fact, a number of tetrads formed by the different nucleobases has been observed in experimental structures (Table 1). They can be classified in two main groups: (a) those formed by arrangements of the same nucleobase (homotetrads), and (b) those formed by the association of two base pairs. The latter can involve canonical G:C or A:T Watson-Crick base pairs, or a variety of mismatches.

## 2. Results

### 2.1. Homotetrads

As in the case of G-tetrads, pure C-[9,10], T-[11,12] and A-tetrads [13] are formed by the arrangement of four identical nucleobases, as shown in Figure 1B–D. These homotetrads have always been observed in the context of parallel G-quadruplex structures, either in internal regions of the quadruplex or as capping elements interacting with terminal G-tetrads. Of particular interest is the case of the oligonucleotide of sequence d (AGAGAGATGGGTGCGTT), which folds as a tetrameric parallel quadruplex. In this case, the four possible DNA homotetrads coexist in the same crystal structure [14]. Homotetrads have also been found in RNA G-quadruplex scaffolds. In this case, U-tetrads (Figure 1E) [15,16,17], usually located at the 3′-end of parallel quadruplexes [18], are the most common. However, RNA homotetrads formed by adenines [19,20] or inosines [21] have also been reported.

The conformation of the T-tetrad is shown in Figure 1D. T-tetrads were first observed in the solution structure of the tetrameric G-quadruplex formed by the sequence d(TGGTGGC) [11]. This tetrad is mainly stabilized by hydrogen bonds between imino protons H(N3) and O4 atoms of adjacent thymines. However, Liu et al. [14] suggested that weak hydrogen bond interactions between H(C5) and O2 atoms may also contribute to the stability. Similar T-tetrads have been observed in the crystal structure of d(TGGGGT)_4_ [22] and in an all-LNA G-quadruplex from 5′-TGGGT-3′. In the latter case, a K^+^ ion bridges the four O4 atoms [23]. Similar hydrogen bonds and K^+^ interactions have been observed in RNA U-tetrads. Theoretical studies confirm the expected higher stability of U-tetrads due to the absence of methyl-O2 repulsions [24].

In the oligonucleotide d(TGGGCGGT), which also adopts a parallel tetrameric G-quadruplex structure, the central cytosines form the C-tetrad shown in Figure 1C [9]. In this structure, amino protons are at a hydrogen-bond distance from the O2 atoms. However, in the C-tetrad formed in the crystal structure of d(AGAGAGATGGGTGCGTT), the interactions between cytosines do not occur by direct hydrogen bonds, but through a network of highly conserved water molecules located in the middle of the tetrad [14]. In the crystal structure of the DNA decamer, d(CCA^CNV^KGCGTGG) (^CNV^K, 3-cyanovinylcarbazole), which forms a G-quadruplex structure in the presence of Ba^2+^, a C-tetrad, is stabilized by water molecules-mediated contacts between the divalent cations and the cytosines, allowing Ba^2+^ ions to occupy the central ion channel [10]. Computational calculation suggests that C-tetrads in the contexts of G-quadruplex have a high propensity for binding alkaline earth cations [25].

A-tetrads are more common and more diverse (Figure 1B). They have been found in a number of DNA and RNA oligonucleotides derived from telomeric sequences, including the NMR structure of human telomere RNA ORN-1 [20], or the structure of (^Br^dU)r(GAGGU) [19]. The A-tetrads observed in RNAs usually form hydrogen bonds between their amino H(N6) and N7 atoms (Figure 1B, right), although N3-H(N6) A-tetrads (Figure 1B, middle) have also been observed in the crystal structure of rU(^Br^dG)r(AGGU) [26]. In the structure of d(AGAGAGATGGGTGCGTT) [14], the hydrogen bonds occur between amino H(N6) and the N3 atoms (Figure 1B, middle), resulting in a bigger central cavity. A third class of A-tetrad, in which hydrogen bonds are formed between H(N6) and N1 atoms (Figure 1B, left), has been reported in the solution structures of d(TGGAGGC)_4_, a sequence that contains the GGAGG repeat present in the C-MYC oncogene [27], the telomeric repeat sequence [28], and the truncated telomeric sequence d(AG_3_T) [13]. In this case, a dynamic exchange is observed between this tetrad and the N7-H(N6) A-tetrad [13]. Although in most A-tetrads adenines glycosidic angles are in anti, the syn conformation is also possible [13].

The overall size of T-, C- and A-tetrads, as estimated from diagonal C1′-C1′ distance, is slightly smaller than in the case of the G-tetrad, being this distance around 1–2 Å shorter in T- and A-tetrads, and ~3 Å shorter in the case of the C-tetrad [14]. Due to their similar size, homotetrads provoke only minor distortions in the structure of parallel G-quadruplexes.

### 2.2. Base-Paired Tetrads

In addition to the homotetrads described above, tetrads can also be formed by the association of two base pairs. In most cases, the base pairs involved are Watson–Crick G:C or A:T base-pairs, although the isomorphous G:T base-pair and other mismatches have also been reported. Base-paired tetrads can be classified in two big groups depending on the relative orientation of the two base pairs: the so-called major groove and minor groove tetrads (Figure 2). Although most of the base-paired tetrads can be classified in these two categories, there are some exceptions of tetrads presenting very unusual nucleobase interactions [29] or water-mediated interactions in complex RNA structures [7].

#### 2.2.1. Major Groove Tetrads

In major groove tetrads, the interaction between base pairs occurs through their major groove side (the base-pair side that would be oriented towards the major groove in a canonical duplex, Figure 2). Among the first cases reported are the major groove G:C:G:C tetrads found in the structure of the fragile X syndrome (CGG)_n_ triplet repeat [16], and in the d(GGGC) repeats of the adeno-associated viral DNA [30]. In both cases, the quadruplex arises from dimerization of two DNA hairpins, stabilized by several G-tetrads and G:C:G:C tetrads. The latter formed by two intra-molecular Watson–Crick base pairs directly facing each other and interacting through intermolecular bifurcated hydrogen bonds between the cytosine H(N4) and the guanine O6 and N7 atoms (Figure 3A). This arrangement gives rise to the so-called “direct” major groove tetrad. These tetrads have also been found in repetitive RNA sequences [31].

In other occasions, the relative position of the two G:C base pairs is shifted with respect to the direct tetrad (Figure 3B). The resulting tetrad has been named as “slipped” major groove tetrad [32]. In this case, the stabilization between base pairs does not arise from additional hydrogen bonds, but through cation coordination with O6(G) and N7(G) atoms from the two guanine residues. Interestingly, the conformations of the G:C:G:C tetrad may depend on the particular cation present in the sample. Thus, the antiparallel quadruplex structure of the d(GGGC)_n_ repeats of the adeno-associated viral DNA in the sodium form contains two direct major groove G:C:G:C tetrads, whereas in the presence of potassium the alignment of the two G:C base pairs is shifted, with the two major groove edges of the guanines coordinating a K^+^ cation [32]. Slipped major groove G:C:G:C tetrads formation has also been reported in a quadruplex-duplex hybrid structure, in which the slipped mixed tetrad is located between the G-quadruplex core and the duplex [33], and in an unusual quadruplex RNA structure formed by the major groove interaction of two antiparallel duplexes [34].

Direct and slipped major groove tetrads can be also formed by the association of A:T Watson-Crick base pairs (Figure 3C,D). The slipped alignment was found in the dimeric solution structure of the octamer d(GAGCAGGT) [35] and implies the formation of two N7(A)-H(N6)(A) additional hydrogen bonds. In the direct alignment, the inter-base pair hydrogen bonds are O4(T)-H(N6)(A). This tetrad has been observed as crystallographic contacts between unit cells in the dimeric propeller-like quadruplex structure of the human telomeric sequence, d(TAGGGTTAGGGT) [1]. Interestingly, these tetrads are not part of the G-quadruplex core but are formed between loop resides, connecting different quadruplexes and contributing to stabilize the crystal.

Major groove tetrads involving mismatch base pairs have also been found. G:T:G:T major groove tetrads (Figure 3E) have been proposed in the structures of some parallel G-quadruplexes [36] and in the interlocked G-quadruplex structure formed by d(GGGT) and derived sequences [37]. G:A:G:A tetrads involving G:A mismatches have been exhaustively explored by DFT methods [38]. These calculations indicate that squared planar G:A:G:A major groove tetrads are not favored, due to repulsion interactions between lone electron pairs of nitrogen atoms. However, an almost planar G:A:G:A tetrad with similar nucleobase orientation has been observed in the solution structure of d(GCGAGGGAGCGAGGG) [39]. This sequence is a variation in the d(GGGAGCG) repeat, found in the regulatory region of the PLEKHG3 human gene related to autism [40]. This structure is a dimer resulting from the self-association of two fold-back loops, and stabilized by a number of unusual elements, such as two G:A:G:A tetrads (Figure 3F) with all their glycosidic angles in anti, G:G base pairs, and direct minor groove G:C:G:C tetrads (Figure 4A).

#### 2.2.2. Minor Groove Tetrads

The association between the two base pairs forming the tetrad can also occur through the minor groove side of the two base pairs (the side that would be oriented towards the minor groove in a canonical duplex, Figure 2). As in the previous case, the resulting minor groove tetrad can be direct or slipped, depending on the relative position of the two base pairs involved. G:C:G:C minor groove tetrads were found in the crystallographic structure of d(GCATGCT) [41,42], and in the solution structures of the cyclic oligonucleotides d<pTGCTCGCT> [43] and d<pCCGTCCGT> [44] (the notation d<p(sequence)> indicates cyclic deoxyoligonucleotide). All these structures are dimers stabilized by intermolecular Watson–Crick base-pairs. In the case of d(GCATGCT) and d<pTGCTCGCT>, the minor groove tetrads are direct, forming two bifurcated H(N2)(G)-O2(C) hydrogen bonds between the base pairs (Figure 4A). In contrast, in the structure of d<pCCGTCCGT> the minor groove tetrads are slipped, forming two H(N2)(G)-N3(G) hydrogen bonds (Figure 4B). No cation sensitivity between these two types of tetrads has been observed. However, the order of residues closing the loops seems to be crucial. Thus, in 5′-C-XX-G-3′ turns the tetrad is direct, and in 5′-G-XX-C-3′ slipped [45].

In most cases, the orientation of two Watson-Crick base pairs forming the tetrads is opposite to each other. However, in the dimeric structures of d(TGCTTCGT) and d(TCGTTGCT), and their cyclic analogue d<pCGCTCCGT> [45], the two G:C base pairs are in the same orientation, giving rise to a C:G:G:C tetrad. In this case, the hydrogen bonds between base pairs are H(N2)(G)-N3(G) and H(N2)(G)-O2(C) (Figure 4C).

Minor groove tetrads can also involve A:T base pairs, giving rise to A:T:A:T tetrads, such as in the dimeric structure of d<pCATTCATT> [43,46], or G:C:A:T mixed tetrads, observed in the dimeric structure of d<pCGCTCATT> [47]. In both cases, only the direct alignment has been observed (Figure 4G,H). A:T:A:T and G:C:A:T minor groove tetrads are stabilized by cation coordination, as shown by crystallographic and theoretical studies.

Likewise, minor groove tetrads can involve one or two G:T mismatches. As in the case of G:C:G:C, the two alignments (direct and slipped) have been observed in G:T:G:T tetrads [48] (Figure 4D,E). In mixed G:C:G:T tetrads, however, only the slipped alignment has been reported (Figure 4F) [48,49]. Minor groove tetrads with G:A mismatches (Figure 4I) have been observed in the dimeric fold-back structure of d(CGTAAGGCGTA) [50]. This structure is stabilized by minor groove G:C:G:C and G:A:G:A tetrads, both in the slipped configurations. The glycosidic angle of adenines in this tetrad is in syn.

In addition to the different hydrogen bonds pattern stabilizing the interaction between the two base pairs, there is a fundamental difference between major and minor groove tetrads, whereas in major groove tetrads, the two base-pairs are co-planar and in minor groove tetrads the two base pairs present a relative inclination that ranges from 20° to 40°, depending on the particular structure. The reason of this effect is not clear. It may be related to the close proximity between phosphate groups in structures stabilized by minor groove tetrads. In these cases, a relative inclination between base pairs may alleviate electrostatic repulsion between phosphates. However, it can also be an intrinsic property of purine-purine interactions through their minor groove side. The structures of parallel DNA duplexes stabilized by homopurine base pairs point towards this possibility, since G:G and A:A base pairs in these duplex structures exhibit a similar inclination [51]. Although a number of theoretical studies have shown that coplanarity between G:C base pairs is the most stable configuration in major groove tetrads [52,53], very little is known from a theoretical point of view about the geometry of minor groove G:C:G:C tetrads.
molecules-27-05287-t001_Table 1Table 1Structures with the different non-G tetrads mentioned in the text with their corresponding references and their PDB codes, when available.TypeTetradCationPDBHomoN1-H(N6) A-1EVM [13], 1NP9 [28]N3-H(N6) ANa^+^6A85 [14], 1MDG [26]N7-H(N6) A-/Na^+^1EVN [13], 1J6S [19]C_4_-/Ba^2+^/Na^+^1EVO [9], 4U92 [10], 6A85 [14]T_4_-/Na^+^/K^+^1EMQ [11], 6A85 [14], 4L0A [23], 1S47 [24] U_4_-1RAU [15], 6GE1 [18], 1J6S [19], 1MDG [26], 1J8G [54], 2AWE [55], 4RKV [56], 4RJ1 [56], 4RNE [56], 4XK0 [57]I_4_K^+^2GRB [21]Base-pairedMajor grooveDirect G:C:G:C-/Na^+^1XCE [29], 1A8N [30], 1A6H [31], 3R1E [34], 1JVC [35], 1NYD [58], 6SYK [59], 6SX6 [59]Slipped G:C:G:CK^+^1A8W [32], 7CV4 [33]Direct A:T:A:T-/K^+^1K8P [1]Slipped A:T:A:T-/Na^+^1XCE [29], 1JVC [35]Direct G:T:G:T-[36,37]Direct G:A:G:A-5M1L, 5M2L [39]Minor grooveDirect G:C:G:C-184D [41], 1MF5 [42], 1EU2 [43], 4ZKK [60], 5FHJ [61]Slipped G:C:G:C-/divalent6MC2, 6MC3, 6MC4, 6N4G [50], 2HK4 [45]Slipped C:G:G:C-2K8Z, 2K90, 2K97 [45]Direct G:T:G:T--[48]Slipped G:T:G:T-1C11 [62], 2LSX [63], 7O5E [64]Slipped G:C:G:T-5OGA [65]Direct A:T:A:TNa^+^1EU6 [43], 284D [46]Direct G:C:A:TNa^+^1N96 [47]Slipped G:A:G:Adivalent6MC2, 6MC3, 6MC4, 6N4G [50]

### 2.3. Structural Context

The different structural features between major and minor groove tetrads have important consequences. First, minor grove tetrads are not easily accommodated within G-tetrads scaffolds. In fact, base-paired tetrads found in the context of G-quadruplex structures are always of the major groove type. Secondly, minor groove tetrads cannot be piled up indefinitely due to the mutual inclination between the two base-pairs (between 20–40°, Figure 5A). No structure with more than two consecutive minor groove tetrads has been reported. Two is probably the maximum number of minor groove tetrads that can be assembled together.

Major groove tetrads are planar and, in their direct configuration, are almost isomorphous with G-tetrads, with diagonal C1′C1′ distance between purines almost identical to G-tetrads, and between pyrimidines around 1 Å shorter. This feature makes direct major groove tetrads perfectly suitable to be stacked between G-tetrads (Figure 5C) and, in fact, they are relatively common in G-quadruplex structures [1,16,30,31,32,33,34,35,36,37,39]. In contrast, slipped major groove tetrads are uncommon and only found in terminal positions [32]. The effect of the presence of G:C:G:C tetrads on G-quadruplexes recognition has been explored [66]. However, very few cases of ligands interacting with non-G tetrads has been reported to date [67,68].

On the other hand, minor groove tetrads have always been found in structures in which no G-tetrad is present. These tetrads appear to induce distinct DNA folding (the term “bi-loop” has been suggested for this motif) [46]. This motif has been usually found in homodimers formed by short linear or cyclic oligonucleotides (Figure 5A), in which dimerization occurs through the formation of two minor groove tetrads with intermolecular base pairs [41,42,43,44,45,46,47,48,49,69].

The tetrads are connected by short loops of one to three residues, with two-residue loops being the most stable [70]. The residues in the first position of the loops stack on top of the tetrads forming a cap at both ends of the structure. A common feature of all these structures is the great proximity of the backbones of the two strands in the same subunit. The very short distances between phosphates provokes unfavorable electrostatic interactions, which are partially alleviated by hydrophobic contacts between deoxyriboses.

This peculiar feature is common to other non-canonical DNA structure, the i-motif [71,72]. The i-motif is a four-stranded structure not stabilized by tetrads, but by intercalation of C:C^+^ base pairs. These structures are formed by the association of two parallel-stranded duplexes through hemiprotonated C:C^+^ base pairs. The two duplexes are intercalated in opposite orientations. Since the C:C^+^ base pair requires partial cytosine protonation, i-motif structures are usually more stable at acidic pH. Interestingly, minor groove tetrads have been observed as capping elements in several i-motif structures. The first case reported was that minor grove G:T:G:T tetrad observed in dimeric structures of centromeric sequences [62]. Then, G:T:G:T [63] and G:C:G:T [65] tetrads were observed in other dimeric and monomeric i-motifs. In all cases, the tetrads are in the slipped configuration, and provoke a dramatic thermal and pH stabilization of the i-motif structure. This stabilization is most probably due to the favorable interaction of the charged C:C^+^ base-pair with the guanines of the tetrad (Figure 5B). The capability of minor groove tetrads to stabilize i-motifs may have profound consequences in biology, since consensus sequences based on these interactions have been found to be prevalent in the human genome, occurring preferentially near regulatory regions [65].

Although several i-motifs have been found to be stable at a neutral pH, to date, the only i-motif structures determined at nearly physiological conditions are structures stabilized by minor groove tetrads. Of particular interest is the structure adopted by oligonucleotides containing several repeats of the sequence 5′-dCCGTTCCGT-3′. At a neutral pH, these sequences fold into i-motif structures stabilized by two C:C^+^ base pairs and two C:G:C:G minor groove tetrads. This structure is peculiar since it contains neutral and hemiprotonated cytosines under the same experimental conditions. C:G:C:G tetrads are not stable at acidic pH, in which conditions cytosines are partially protonated. When lowering the pH from 7 to 5, these oligonucleotides undergo a large conformational transition towards a different i-motif structure, with four C:C^+^ base pairs and capped by two G:T:G:T tetrads [73]. This conformational transition was further used to explore the use of a fluorescent cytosine analogue tricyclic 1,3-diaza-2-oxophenoxazine (tC^O^) as an efficient internal probe. Interestingly, the increased stacking interactions between tC^O^ and the guanine residue of the tetrad, when located in contiguous positions, dramatically enhanced the thermal and pH stability of the i-motif structures [74].

Very recently, minor groove tetrads have been used to stabilize DNA constructs, in which i-motif and B-DNA coexist in the same structure, forming i-motif/duplex junctions [64]. These constructs consist of an i-motif moiety capped by a minor groove G:T:G:T tetrad at one of its ends, and an stem-loop hairpin at the other end of the i-motif. Stabilization conferred by the minor groove tetrad is essential to stabilize the structure at neutral pH, since analogous constructs with sequences unable to form the tetrad are not stable [64].

### 2.4. Base Paired Tetrads in Higher-Order Structures

In some occasions, major groove G:C:G:C tetrads are found in structures containing multiple interlocked G-quadruplexes in the regions connecting different units [58,75]. The ability to interlock multiple G-quadruplexes through these tetrads has been used to build nanowires [69,76,77]. In the case of sequences containing 5′-GC-3′ in the terminal position, this assembly of multimeric G-quadruplexes is cation dependent [59]. Similar interlocked structures stabilized by major groove G:C:G:C tetrads have been also found in RNA G-quadruplexes [16]. Inclusion of G:C:G:C tetrads in G-quadruplex scaffolds can be also used for avoiding the inherent polymorphism of these structures when designing DNA nanoassemblies [78]. Furthermore, the formation of G:C:G:C tetrads has been proposed in higher order structures formed by multiple GGGGCC repeats found in the sequence of the C9ORF72 gene [79]. Expansion of these repeats is considered the most important genetic cause of the Amyotrophic Lateral Sclerosis (ALS).

On the other hand, G:C:G:C tetrads, usually in their minor groove conformation, are common between symmetry-related molecules in crystallographic structures. On many occasions, they are formed between terminal G:C base pairs in the crystal structures of DNA duplexes, contributing to stable crystal packing [80]. Slipped tetrads found in the context of a crystallographic network are more common between DNA duplexes that crystallize in B-form [81,82]. In contrast, direct minor groove tetrads are found in A-form DNA structures. The direct arrangement may be facilitated by the wider minor-groove characteristic of A-form double helices [83]. Although not so common, these interactions do not involve terminal residues. This is the case of the DNA-RNA chimeric duplex of sequence d(CCGGC)r(G)d(CCGG). Interestingly, direct minor grove G:C:G:C tetrads connecting different duplexes involve the single 2′-hydroxyl group per strand. This result indicates that minor groove tetrads can be formed with at least one of the four nucleotides being ribo-. However, studies on the association between cyclic and linear oligonucleotides suggest that minor groove tetrads containing ribonucleotides are much less stable than those formed by four dexoyribonucleotides [49].

Minor groove G:C:G:C tetrads have also been found involving terminal G and C residues that do not form base-pairs with their own duplex but with symmetry related ones, forming junction-like quadruplexes. Such structures have been found in pure DNA crystals and in several bisintercalative complexes involving acridine derivatives. In these complexes, the acridine derivatives do not intercalate in the usual way, but interact with the terminal nucleotides of four DNA duplexes forming a large intercalation platform between two G:C:G:C tetrads [84,85,86,87,88]. The structures of these complexes have attracted significant interest because some of the intercalating drugs involved in them are potent topoisomerase inhibitors. Interestingly, in most cases, these structures have been found in crystals containing Co^2+^ ions. Such counterions might be relevant to increase the stability of minor groove tetrads, since cobalt hexamine residues have also been found to stabilize the crystal lattice in one of the crystallographic structures of the dimeric linear heptamer d(GCATGCT) [41,42,60,61].

Base-paired tetrads have also been observed in ordered nanostructures based on G:C pairing. High resolution scanning tunnelling microscopy studies in adlayers formed by coadsorption of guanine and cytosine at Au(111) [89], and at graphite surfaces [90] have revealed the formation of well-ordered periodic structures based on G:C:G:C tetrads in the solid/liquid interface. Such structures are formed by alternate arrangements of G:C base pairs though their major and minor groove sides. Similar supramolecular nanopatterns based on A:T:A:T have also been reported [90]. More recently, base paired C:G:G:C, G:C:G:C and A:U:A:U tetrads have been used as building blocks for the formation of 2D-nanoporous networks [91]. Finally, G-quartets and other homotetrads have been also found in template-assembled synthetic supramolecular structures [92,93,94].

## 3. Conclusions

In summary, the number of arrangements of four nucleobases found in experimental structures amply exceeds the canonical G-tetrads. In this review we have classified these non-guanine tetrads in two main groups: homotetrads, formed by the arrangement of identical nucleobase, and base-paired tetrads, formed by the association of two base pairs (G:C, A:T or mismatches). Most of the tetrads in the second group can be classified in major groove or minor groove tetrads, depending on the side of the two base pairs interacting with each other. The structural contexts in which these tetrads occur is conditioned by their specific geometry. Thus, whereas homo- and major groove tetrads are mainly formed in the context of G-quadruplex structures, minor groove tetrads occur in i-motifs and, in peculiar, fold-back DNA structures. Interestingly, non-canonical tetrads are involved in different DNA-DNA recognition events, such as loop-loop association, formation of G-quadruplex assemblies, or DNA duplex packing. These tetrad-mediated interactions may be involved in biological processes and may have useful applications in DNA nanotechnology.

## Figures and Tables

**Figure 1 molecules-27-05287-f001:**
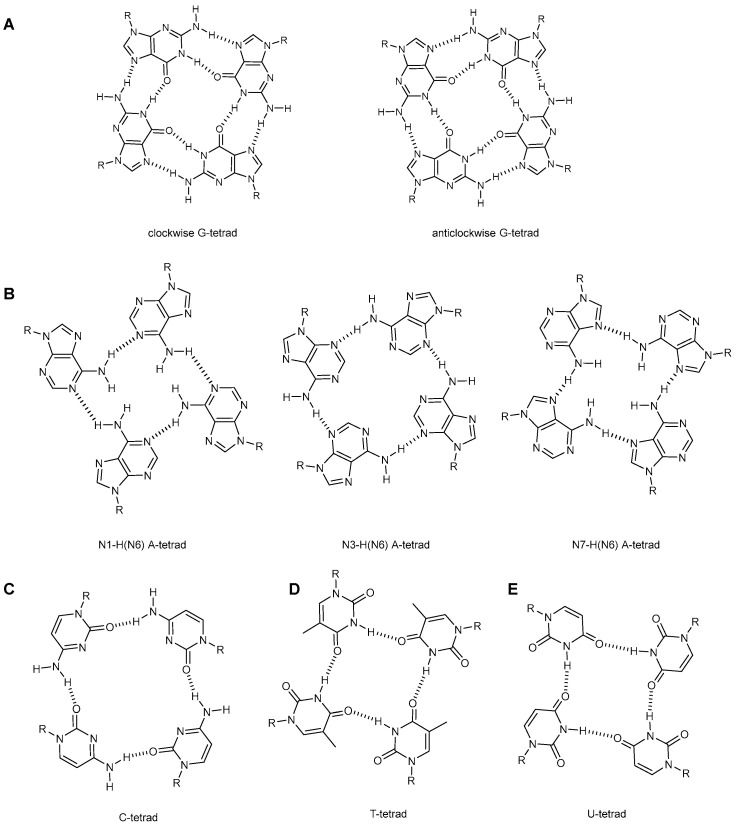
Schemes of the different homotetrads discussed in the text. (**A**) G-tetrads in their two possible orientations (clockwise and anticlokwise); (**B**) The three most common A-tetrads; (**C**–**E**) C-, T-, and U-homotetrads, respectively.

**Figure 2 molecules-27-05287-f002:**
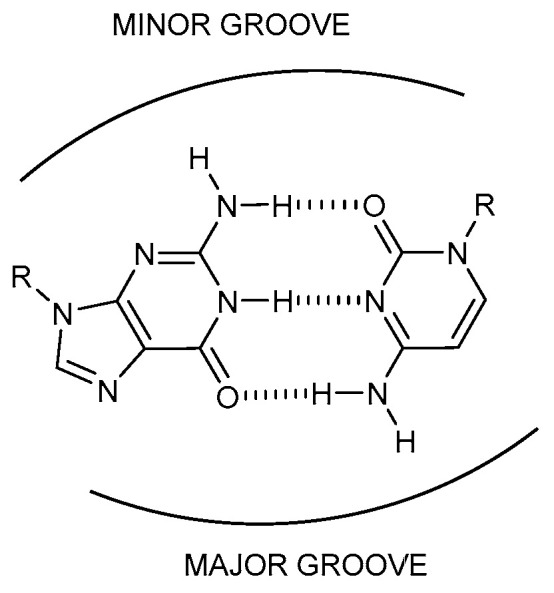
Major and minor groove sides of a G:C Watson-Crick base pair.

**Figure 3 molecules-27-05287-f003:**
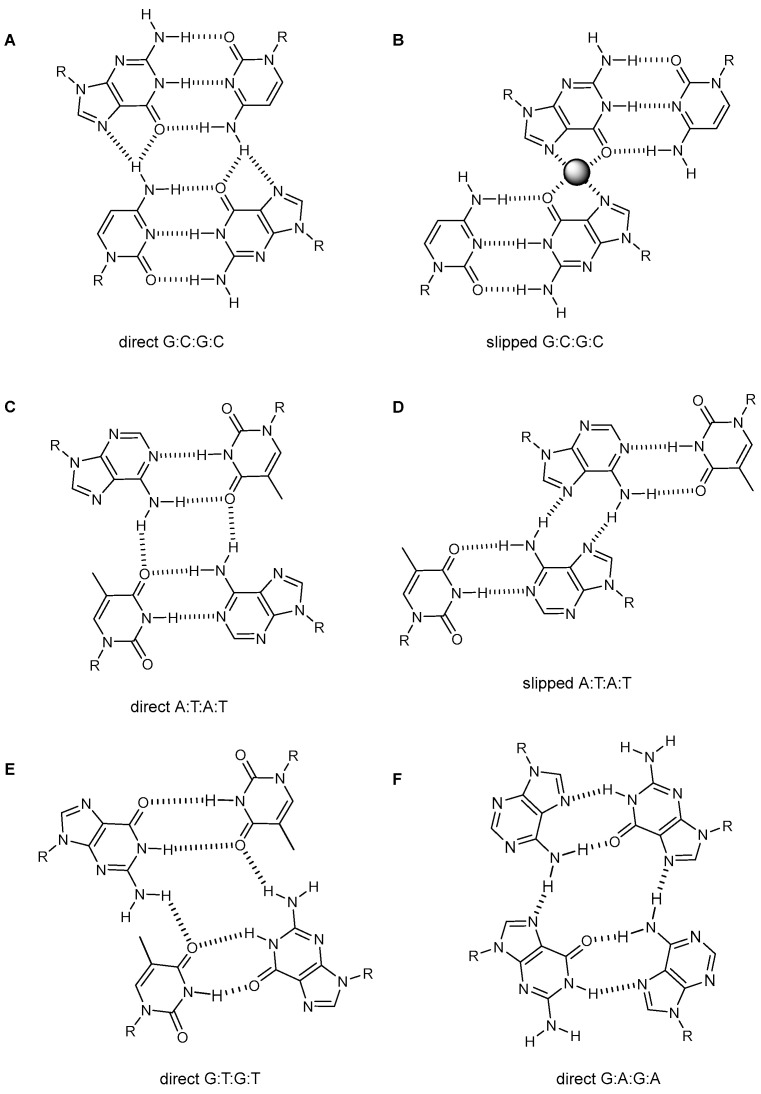
Schemes of the different major groove tetrads discussed in the text. (**A**) Direct major groove G:C:G:C tetrad; (**B**) Slipped major groove G:C:G:C tetrad; (**C**) Direct major groove A:T:A:T tetrad; (**D**) Slipped major groove G:C:G:C tetrad. (**E**) Direct major groove G:T:C:T. (**F**) Direct major groove G:A:G:A tetrad.

**Figure 4 molecules-27-05287-f004:**
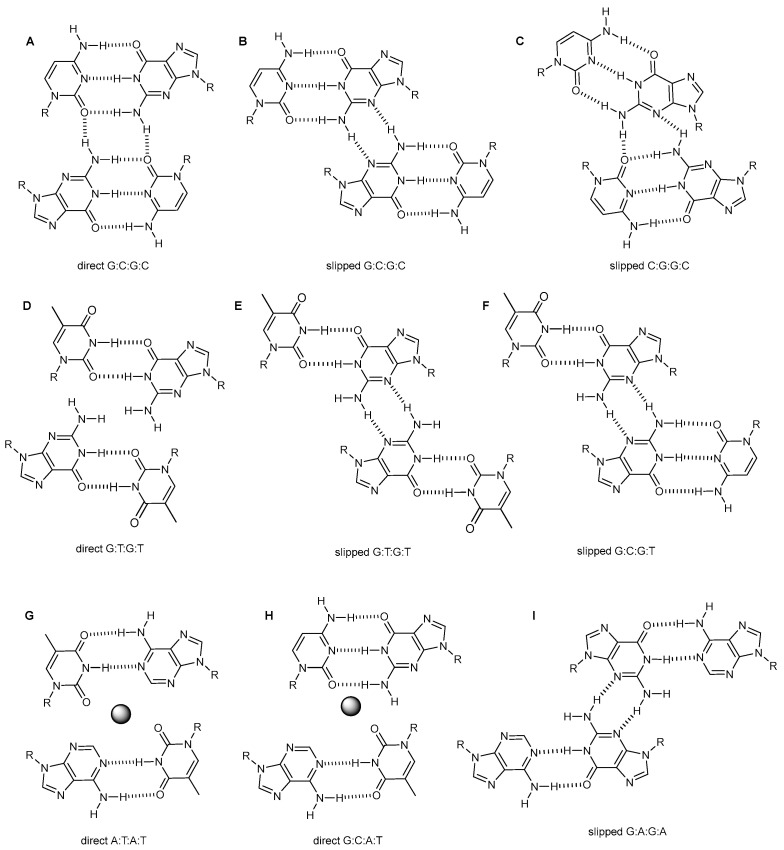
Schemes of the different minor groove tetrads discussed in the text. Direct (**A**) and slipped (**B**) minor groove G:C:G:C tetrads. (**C**) Minor groove C:G:G:C tetrad. Direct (**D**) and slipped (**E**) minor groove G:T:G:T tetrads. (**F**) Slipped minor groove G:C:G:T. Direct minor groove A:T:A:T (**G**) and G:C:A:T (**H**) tetrads stabilized by cation coordination (dark circle). (**I**) Slipped minor groove G:A:G:A tetrad.

**Figure 5 molecules-27-05287-f005:**
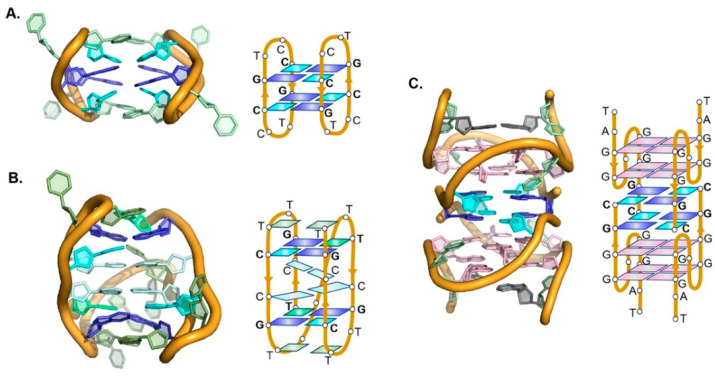
Examples of structures containing non-G tetrads. (**A**) Dimeric structures formed by two cyclic oligonucleotides <dCCGTCCGT> stabilized by two minor groove C:G:C:G tetrads (PDB 2HK4). (**B**) Minimal i-motif structure stabilized by two C:C^+^ base pairs capped by two minor groove G:C:G:T tetrads (PDB 5OGA). (**C**) Association of two G-quadruplexes through formation of two major groove C:G:C:G tetrads (PDB 1NYD).

## Data Availability

Not applicable.

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
