# Peer review of "Non-G Base Tetrads"

_molecules, 2022, doi:10.3390/molecules27165287_

Round 1

Reviewer 1 Report

This manuscript deals with a nice and useful review of the possible non-G base tetrads. This review is interesting because four-stranded structures are attracting an enormous attention in the last few years. Most of the four-stranded structures contains tetrads formed exclusively by G bases.  However, several tetrads formed by the different nucleobases have been observed experimentally. Thus, this review reports the diversity of these non-canonical tetrads, and the structural context in which they have been found. The review deserves to be published as it is.

Author Response

We appreciate the referee's effort in reviewing our mansucript. Thank you very much for the nice evaluation.

Reviewer 2 Report

The review paper “Non-G base tetrads” by Núria Escaja1, Bartomeu Mir, Miguel Garavís and Carlos González reports the structure of a number of non-G tetrads formed by the different nucleobases. The non-G tetrads are classified in two main groups, homotetrads and base-paired tetrads. The base paired tetrads are classified in two big groups, major groove tetrads and minor groove tetrads. The present review paper is written well, but some corrections are necessary.  On the basis of these considerations, I have to conclude that the paper is to justify publication in the journal of Molecules after the following some corrections.

Line 8 in the second paragraph of 1. Introduction in page 1: The authors write that “by electrostatic interactions with monovalent cations located between two consecutive tetrads”. However, a part of G-quadruplexes, such as TBA [d(GGTTGGTGTGGTTGG)] and PS2.M [d(GTGGGTAGGGCGGGTTGG)], are stabilized by a divalent lead cation. The authors should describe that a part of divalent cations play an important role in the stabilization of G-quadruplexes.  

Line 6 in the first paragraph of 2. Results in page 2: “the four possible DNA homotetrads, A-, G-, C-, and T-tetrads,” may be better.

Lines 2 and 3 in page 3, and line 5 from the bottom of page 8: “Angstrom” should be changed into SI unit “nm”. 1 Angstrom is equivalent to 0.1 nm.

Line 2 in the third paragraph of page 4: “Figure 3C and 3D” is correct.

Legend of Figure 3 in page 5: “D) Slipped major groove A:T:A:T tetrad” is correct. “E) Direct major groove G:T:G:T tetrad” is correct. “F) Direct major groove G:A:G:A tetrad” may be better.

Line 4 in the first paragraph of page 5: “G:A:G:A tetrads (Figure 3F)” may be better.

Line 2 from the bottom of page 5: “G:C:G:C tetrads (Figure 4A)” may be better.

Line 3 of page 6: “in a canonical duplex, see Figure 2” may be better.

Line 7 of page 6: General readers are not familiar with the description of “d<p(base sequence)>”. What does “<p” and “>” mean? Some explanation may be necessary.

Line 4 in the 4th paragraph of page 6: “with G:A mismatches (Figure 4I)” may be better.

Table 1 in page 8: What does the number in bracket [] mean?  Is this a reference number? Its explanation is necessary.

Line 7 in “Homo”, line 6 in “Major groove”, and line 4 in “Minor groove” in Table 1 in page 8: PDB numbers are missing. The reason for the missing should be described.

Figure 5 in page 9: The title of this figure is missing. The main concept of this figure is unclear.

Figure 5B and Figure 5C in page 9: These figures are not cited in the main text. Are these figures necessary? The necessity of these figures should be described.
